# Effects of SKCPT on Osteoarthritis in Beagle Meniscectomy and Cranial Cruciate Ligament Transection Models

**DOI:** 10.3390/ijms241914972

**Published:** 2023-10-07

**Authors:** Hye-Min Kim, Minseok Kang, Yoon-Seok Jung, Yoon-Jung Lee, Wonjae Choi, Hunseung Yoo, JeongHoon Kim, Hyo-Jin An

**Affiliations:** 1Department of Oriental Pharmaceutical Science, College of Pharmacy, Kyung Hee University, Seoul 02447, Republic of Korea; mins7576@daum.net; 2Life Science R&D Center, SK Chemicals, 310 Pangyo, Seongnam 13494, Republic of Korea; minsogy@sk.com (M.K.); tjrvtjr@sk.com (Y.-S.J.); yoonjung@sk.com (Y.-J.L.); atomi@sk.com (W.C.); hs.yoo@sk.com (H.Y.); jeonghoon.kim@sk.com (J.K.); 3Department of Integrated Drug Development and Natural Products, Graduate School, Kyung Hee University, Seoul 02447, Republic of Korea

**Keywords:** osteoarthritis, SKCPT, beagle, meniscectomy, cranial cruciate ligament transection, drug delivery systems

## Abstract

Osteoarthritis (OA) affects >500 million people globally, and this number is expected to increase. OA management primarily focuses on symptom alleviation, using non-steroidal anti-inflammatory drugs, including Celecoxib. However, such medication has serious side effects, emphasizing the need for disease-specific treatment. The meniscectomy and cranial cruciate ligament transection (CCLx)-treated beagle dog was used to investigate the efficacy of a modified-release formulation of SKI306X (SKCPT) from *Clematis mandshurica*, *Prunella vulgaris*, and *Trichosanthes kirilowii* in managing arthritis. SKCPT’s anti-inflammatory and analgesic properties have been assessed via stifle circumference, gait, incapacitance, histopathology, and ELISA tests. The different SKCPT concentrations and formulations also affected the outcome. SKCPT improved the gait, histopathological, and ELISA OA assessment parameters compared to the control group. Pro-inflammatory cytokines and matrix metalloproteinases were significantly lower in the SKCPT-treated groups than in the control group. This study found that SKCPT reduces arthritic lesions and improves abnormal gait. The 300 mg modified-release formulation was more efficacious than others, suggesting a promising approach for managing OA symptoms and addressing disease pathogenesis. A high active ingredient level and a release pattern make this formulation effective for twice-daily arthritis treatment.

## 1. Introduction

Osteoarthritis (OA) is the most prevalent joint disease, resulting in significant disability among older adults. OA is a debilitating disease associated with pain, joint stiffness, and muscle atrophy, which is characterized by cartilage loss, bone remodeling, and, often, synovial inflammation triggered by an imbalance in joint anabolic and catabolic processes [1]. In 2021, >22% of individuals aged ≥40 years were diagnosed with knee OA, with current estimates suggesting that >500 million people globally suffer from OA. According to a substantial cohort study conducted within the United States, the prevalence of knee OA has increased by 2.1-fold since the 1950s. By 2032, the prevalence rate of OA is projected to escalate from 26.6% to 29.5%. This widespread condition results in high costs and mortality rates [2,3,4]. There is no definitive cure for OA, and treatment research is ongoing. For a considerable duration, OA’s clinical interventions prioritized pain relief over halting disease progression [5]. However, a recent shift in OA management strategies emphasizes early prevention and an effort to arrest or decelerate the disease’s progression before severe damage transpires [6]. Consequently, it is vitally important to thoroughly understand and identify potential biomarkers and therapeutic targets throughout the various stages of OA progression [1].

The management of OA primarily addresses symptoms rather than the underlying cause, with non-steroidal anti-inflammatory drugs (NSAIDs) representing the central treatment approach. Celecoxib, an NSAID, is a representative selective cyclooxygenase-2 (COX-2) inhibitor administered to alleviate pain and inflammation associated with arthritis [7]. However, NSAIDs carry significant side effects, such as gastrointestinal irritation, bleeding, and renal impairment, necessitating the consideration of alternative treatments for patients with these associated risk factors [8]. When using Celecoxib to treat OA, potential side effects include gastrointestinal problems, cardiovascular risks, edema, kidney issues, allergic reactions, liver function impact, gastrointestinal bleeding, headache, dizziness, and an increased risk of blood clots [9,10]. Because existing treatment options mainly target pain relief, researchers are exploring novel therapeutic strategies to address the disease’s root cause [11]. As part of this strategy, SKI306X is designed to mitigate the systemic side effects of anti-inflammatory analgesics, offering a holistic approach that includes preventing joint tissue damage, safeguarding existing structures, and enhancing tissue regeneration [12].

SKCPT is a modified-release formulation of SKI306X, an active pharmaceutical ingredient derived from the traditional herbs *Clematis mandshurica*, *Prunella vulgaris*, and *Trichosanthes kirilowii*, which has been shown to treat arthritis. [13]. SKCPT is a medicine that maintains the daily dosage of SKI306X while adjusting the dose to improve ease of administration and achieves formulation improvements through controlled-release mechanisms. According to the findings of previous studies, SKI306X influences various receptors, including GABA, opiate, thromboxane A2, serotonin, and nicotinic acetylcholine, hinting at its potential analgesic mechanisms [6]. In in vitro models, SKI306X and its components inhibit pro-inflammatory cytokines and reduce IL-1b-induced proteoglycan (PG) degradation and nitric oxide production in LPS-stimulated human PBMC and IL-1b-stimulated bovine cartilage explants [6,14]. SKI306X inhibits PGE2 production by suppressing COX-2 expression in murine macrophages and LTB4 production by inhibiting 5-LOX activity in human whole blood [15]. Furthermore, in rabbit OA models, SKI306X has demonstrated the significant inhibition of collagenase-induced OA progression and protection against IL-1-induced PG degradation and induced apoptosis in cartilage [13,16]. It also mitigated the IL-1α-stimulated increase in the expressions of MMP-3 and MMP-13, critical enzymes in collagen degradation, in rabbit cartilage cultures [12,16]. Additionally, placebo-controlled clinical studies demonstrated the beneficial effects on patients with classical OA of the knee, with significant improvements in the visual analog scale, the Lequesne index, and patients’ and investigators’ opinion, with comparable efficacy to diclofenac but fewer side effects [17]. Since SKCPT has the same herbal composition as SKI306X, it is expected to yield similar results as those previously described in terms of efficacy. Moreover, active ingredients in SKCPT, including oleanolic acid, rosmarinic acid, and rutin, exhibit multifunctional properties such as anti-inflammation and enhancement of blood circulation [18].

Despite the heterogeneity of OA, large animal models, especially dogs, are preferred due to their superior anatomical resemblance to humans, both macroscopically and microscopically, thus yielding more clinically relevant data. Histological and biochemical properties of articular cartilage, subchondral bone, synovium, joint capsule, and menisci are well conserved between dogs and humans [19,20,21]. Prompted by the high incidence of OA in canines, diagnostic monitoring protocols have been instituted. However, considering the significant anatomical and pathological differences among species, histological examination remains the benchmark for outcome evaluation in canine OA models [22,23]. The Osteoarthritis Research Society International (OARSI) dog working group further emphasizes this, underscoring that, despite the strides made in non-invasive monitoring, histological examination remains the primary method for assessing the severity and extent of pathology in OA models [24]. Research guidelines advise using skeletally mature dogs with no spontaneous joint pathology for studies, avoiding certain breeds to limit potential confounding variables, thus making the representative beagle model a suitable candidate under these conditions [20,25]. As part of this, the meniscectomy and cranial cruciate ligament transection (CCLx)-treated beagle dog was used in this experiment. The cranial cruciate ligament (CCL) in dogs primarily serves as the principal restraint against anterior tibial displacement relative to the femur and safeguards against stifle joint hyperextension; its insufficiency results in modified joint biomechanics, heightened femorotibial cartilage stresses, and the rapid onset and progression of OA, along with an elevated vulnerability to secondary meniscal injuries [26,27].

As previously outlined, numerous studies have indicated the effectiveness of SKI306X in managing arthritis for clinical treatment. Nevertheless, the precise effectiveness of the drug in the meniscectomy and CCLx model, leveraging the beagle dog, a prominent model for OA, has yet to be determined. Moreover, there is a lack of comprehensive experimentation assessing the impacts of different concentrations and formulations of SKI306X. Therefore, this study aimed to shed light on these aspects. In this study, the efficacy of the SKCPT tablet was examined to minimize the shortcomings of SKI306X and maximize its advantages.

## 2. Results

### 2.1. The Significance of the Tablets Was Confirmed through Dissolution Test

The dissolution rate was determined by measuring the content of rosmarinic acid, an active component of herbal extracts. The quantification of the rosmarinic acid content followed a methodology akin to prior studies [28]. In Figure 1, guided by the preceding evaluations, four formulations were selected for dissolution testing: 200 mg immediate-release (IR) tablet, 300 mg IR tablet, 300 mg modified-release (MR) tablet, and 300 mg controlled-release (CR) tablet. The rosmarinic acid dissolution results revealed that the point at which the average dissolution rate reached 80% occurred at 30 min for ‘200 mg IR tablet’, 45 min for ‘300 mg IR tablet’, 90 min for ‘300 mg MR tablet’, and 360 min for ‘300 mg CR tablet’. Each formulation demonstrated IR, MR, and CR characteristics, respectively.

### 2.2. Identification of Consistency for Comparison between Groups

Upon measuring body weights, no notable discrepancies in statistical significance were observed between the various test groups throughout the entire duration of the experiment (Figure 2A). This observation indicates the preservation of similar physiological conditions across all animal cohorts, as inferred from consistently similar appetite patterns, physical activity, and metabolic processes. Regarding the measurements of the stifle circumference (Figure 2B), the extent of joint edema in G2, G3, G5, G6, and G7 was significantly greater than that in the normal control group 14 days before the test substance administration. This phenomenon is postulated to be a temporary manifestation of joint edema consequential to the induction of arthritis through surgical means. Although the increase in swelling was not statistically significant, it was still evident that the level of edema in the arthritis-induced groups was higher than that of the normal group.

### 2.3. SKCPT Treatment Improved the Most Kinetics Assessment

Concerning the kinetics assessment (Figure 3), the gait scores from day 1 before the start of the test substance administration to the end of the test (Day 56) revealed that all arthritis-induced groups (G2–7) had statistically significantly higher gait assessment scores compared to G1 (*p* < 0.05). Furthermore, on the 19th day following the commencement of tablet administration, the gait assessment scores in G4 and G6 were significantly lower than those in G2 (*p* < 0.05). Similarly, on the 26th, 28th, and 33rd day after the initiation of the tablet administration, the gait assessment score levels in G3, G4, and G6 were statistically significantly lower than in G2 (*p* < 0.05).

### 2.4. SKCPT Treatment Significantly Improved the Right Hind Limb Weight-Bearing Capacity

Following the execution of the incapacitance test to evaluate changes in limb buoyancy (Figure 4), it was observed that the right hind limb weight-bearing ratio in groups G2–G7 from the pre-administration period to the 42nd day post-administration was significantly diminished relative to G1 (*p* < 0.001, *p* < 0.01, or *p* < 0.05). On the 49th day post-initiation of the administration, the right hind limb weight-bearing capacity levels of G2, G5, and G7 were also significantly lower than those of G1 (*p* < 0.01). By the end of the experiment period, the right hind limb weight-bearing capacity levels of G2, G3, G4, G5, and G7 were significantly lower than those of G1 (*p* < 0.001, *p* < 0.01, or *p* < 0.05). In contrast, the right hind limb weight-bearing ratios of G4 and G6 were significantly superior to G2 (*p* < 0.01 or *p* < 0.05). Additionally, the capacity of G6 was statistically significantly higher than that of G5 (*p* < 0.05), whereas G7 displayed a right hind limb weight-bearing capacity level significantly lower than G6.

### 2.5. SKCPT Alleviated the Images of OA Severity and OARSI Scores

Through histological examination, the total cartilage score, which quantifies the degree of cartilage damage, and the total synovial score, which evaluates the structure of the synovial tissue along with the escalation of inflammatory cells, vascular dilation, and tissue swelling, were assessed via hematoxylin and eosin (H&E) (Figure 5A and Appendix A) and Safranin O staining (Figure 5B). Additionally, the OARSI score was utilized to evaluate the severity of OA through the aforementioned staining methods (Figure 5C). Compared to group G1, the OARSI score levels of groups G2, G3, G4, G5, and G7 were statistically and significantly elevated (*p* < 0.05). Conversely, the OARSI score levels of groups G4 and G6 were significantly reduced relative to group G2 (*p* < 0.05).

### 2.6. SKCPT Reduced the Expression of OA-Related Proteins

Herein, immunohistochemistry (IHC) staining was detected to examine the expression levels of matrix metalloproteinase 13 (MMP13), tissue inhibitor of metalloproteinases 1 (TIMP1), and collagen type II (Col-II). MMP13 is noted for its role in tissue damage and regeneration, while TIMP1 is an inhibitor of metalloproteases [29,30]. Collagen II is a predominant component of connective tissue typically found within cartilage tissue [31]. Upon assessment of the MMP-13 positive area (%), the level in group G2 was found to be significantly higher than that in group G1 (*p* < 0.01). Moreover, the level of the MMP-13 positive area (%) in group G6 was statistically lower than that in group G2 (*p* < 0.05) (Figure 6A,D). In the evaluation of the TIMP-1 positive area (%), group G2’s level was statistically higher than that of group G1 (*p* < 0.01), whereas the levels in groups G4 and G6 were significantly lower compared to group G2 (*p* < 0.05) (Figure 6B,E). Upon evaluating the collagen type II (Col-II) positive area (%), no statistically significant differences were discerned among all the test groups (Figure 6C,F).

### 2.7. SKCPT Significantly Reduced the OA-Related Inflammatory Markers

As a result of the ELISA analysis, the levels of IL-1β in G2, G3, G4, G5, and G7 were statistically much higher than those in G1 (*p* < 0.001 or *p* < 0.01). The IL-1β levels in G3, G4, G5, G6, and G7 were significantly lower than those in G2 (*p* < 0.001 or *p* < 0.01), and the IL-1β levels in G6 were statistically lower than those in G5 (*p* < 0.05), while the level of IL-1β in G7 was statistically higher than that in G6 (*p* < 0.01) (Figure 7A). For MMP-3, a significant increase was observed in the MMP-3 levels of G2, G3, G4, G5, and G7 when compared to group G1 (*p* < 0.001, *p* < 0.01, or *p* < 0.05). In addition, G3, G4, G5, G6, and G7 showcased significantly lower MMP-3 concentrations than group G2 (*p* < 0.001, *p* < 0.01, or *p* < 0.05). Specifically, the MMP-3 concentration in G7 was found to be statistically higher than that in G6 (*p* < 0.05) (Figure 7B). In the case of TNF-α, the TNF-α levels in G2, G3, G4, G5, and G7 were significantly higher than those in G1 (*p* < 0.01 or *p* < 0.05), and the TNF-α levels in G3, G4, G5, and G6 were significantly lower than those in G2 (*p* < 0.001). Notably, group G5 exhibited a significantly higher TNF-α concentration than G6 (*p* < 0.05), while group G7 displayed a significantly higher TNF-α level compared to groups G3, G4, and G6 (*p* < 0.001 or *p* < 0.01) (Figure 7C).

## 3. Discussion

OA is a degenerative condition characterized by progressive joint cartilage deterioration, manifesting as pain and stiffness in the joints [32]. Multiple factors, including advanced age, genetic predisposition, obesity, previous joint injuries, and repeated joint use, contribute to the etiology of OA [2,33]. Pertinently, cytokines are fundamental to the pathophysiology of OA, given their role in inciting inflammation and promoting the degeneration of cartilage, thus intensifying the progression of the disease [34,35]. Pro-inflammatory cytokines, notably IL-1 and TNF-a, are central in initiating and developing OA [36]. These cytokines, produced by mononuclear cells, synoviocytes, and chondrocytes, stimulate the production of additional inflammatory mediators like IL-6, IL-8, IL-18, PGE2, nitric oxide (NO), and matrix metalloproteinases (MMPs). Cartilage affected by OA is a significant production site for mediators and cytokines associated with inflammation, which activate chondrocytes into a catabolic condition, contributing to a matrix-degrading activity, including protease secretion, radical species production, down-regulation of matrix and protease inhibitor synthesis, inhibition of chondrocyte proliferation, and cell death. Therefore, inhibiting these inflammatory mediators could be a key strategy in treating OA [7,11,34]. IL-1β, a primary pro-inflammatory cytokine implicated in OA, activates numerous signaling pathways leading to OA progression. However, inhibiting IL-1β has not yielded the expected halt in OA progression, suggesting many factors beyond a single cytokine influence the disease’s pathogenesis. TNF-α works in synergy with IL-1β, contributing significantly to the pathogenesis of OA as a potent pro-inflammatory cytokine instrumental in cell differentiation, proliferation, and apoptosis. TNF-α also hinders chondrocytes from synthesizing PG components and type II collagen while promoting ECM degradation via the induction of collagenases (MMP-1, MMP-3, MMP-13) and aggrecanases [29,30,31].

This study implemented animal trials that involved the partitioning of the drug SKI306X as the most effective pharmaceutical active ingredient associated with diverse drug delivery systems, namely immediate release (IR), modified release (MR), and controlled release (CR). Each delivery mechanism uniquely facilitates the drug’s release, moderating the absorption duration [37]. The selection of a particular active ingredient is predicated upon the requirements and objectives of specific scenarios, with key considerations such as the longevity of the drug’s impact, the administration frequency, and the absorption rate guiding its therapeutic application [38]. Traditional IR rapidly disperses the drug, which may necessitate frequent administration. MR moderates drug release to maintain a continuous effect, thereby reducing dosing frequency. CR ensures steady effects by controlling drug release and maintaining a stable plasma concentration [8,39,40]. The assumption underlying this experiment was that altering the release rate of a drug containing the same active ingredient would control its absorption rate. To prevent a potential decrease in drug absorption at high doses and improve medication adherence, we sought to reduce the administration frequency by modifying the release rate. As a result, the 300 mg MR formulation, SKCPT, demonstrated improved effects compared to the same high dose of 300 mg IR, confirming its efficacy. In this investigation, SKCPT exhibited the most pronounced efficacy when formulated as the MR.

Beagle dogs were used as models for medial meniscectomy and CCLx under the stipulated experimental conditions. Upon the administration of the test substance, a tendency for improvement was observed in the gait evaluation incapacitance test, histopathological examination, and ELISA analysis compared to the control group. Notably, the group treated with 300 mg of MR (SKCPT) exhibited superior efficacy. The groups in the experiment were as follows: normal control (G1), vehicle control (G2), positive control (G3), 200 mg IR (G4), 300 mg IR (G5), 300 mg MR (G6, SKCPT), and CR-1 (G7). Throughout the experimental period, no significant weight changes were noted. Joint swelling was higher in the test group than in the normal group, likely due to temporary inflammation caused by arthritis-inducing surgery. The arthritis-induced groups demonstrated higher scores in gait assessment than the normal group, and some others exhibited lower scores after substance administration. The right hind limb weight-bearing ratio was lower in the arthritis-induced group compared to the normal group, and some other groups maintained a low level even after administration. Certain test groups tended to have a higher or lower right hind limb weight-bearing ratio. Regarding histopathological examination, most groups showed lower OARSI scores than the vehicle group, except for the CR-1 group. IHC staining of tissues demonstrated that the positive areas of MMP-13 and TIMP-1 were higher in the administered group compared to the normal group. They were lower in all groups compared to the control group, with the group administered with 300 mg of MR (SKCPT) being the lowest. The positive area of TIMP-1 was significantly lower in the group administered with 200 mg of IR and 300 mg of MR (SKCPT). The positive area of Col-II showed a higher tendency in the administered groups than the control group, though no significant difference was observed.

In the results of the ELISA tests, TNF-α, IL-1β, and MMP-3 expressions were higher in all the treated groups compared to the normal group. The positive control group, 200 mg IR-treated group, 300 mg IR-treated group, 300 mg MR (SKCPT)-treated group, and CR-1-treated group had lower levels than the control group. The expression of TNF-α was higher in the 300 mg IR-treated group than in the 300 mg MR (SKCPT)-treated group, and the CR-1-treated group was higher than the positive control group, 200 mg IR-treated group, and 300 mg MR (SKCPT)-treated group. The expression of MMP-3 was higher in the CR-1-treated group than in the 300 mg MR (SKCPT)-treated group. Based on the above results, it is hypothesized that the therapeutic effect of the 300 mg MR (SKCPT)-administered group is the most remarkable, particularly demonstrating a significant level of arthritis improvement in the incapacitance test and ELISA categories when compared to CR-1. Furthermore, in the case of the 300 mg MR (SKCPT)-administered group, it appears to possess equal or superior therapeutic effects on arthritis improvement compared to the positive control group and 200 mg IR-administered group. From a formulation perspective, SKCPT administration resulted in a reduction in arthritis lesions and an improvement in abnormal gait caused by arthritis. The efficacy of the 300 mg MR (SKCPT) formulation was found to be superior to that of the CR-1 formulation. Additionally, it exhibited comparable or even greater efficacy similar to the positive control substance, and the 200 mg IR formulation administered thrice daily.

## 4. Materials and Methods

### 4.1. Animals

All animal experiments were approved by the Institutional Animal Care and Use Committee of Knotus (No. IACUC 21-KE-199). Thirty-one male non-naïve beagle dogs (12–17 months) were obtained from Xi’an Dilepu Biology & Medicine Co., Ltd. (Lianhu, China). During a 10-day acclimatization period following the acquisition, general symptoms were monitored to verify their health status, and 28 healthy animals were selected for the experiment. The animals were identified by marking their ears with a meteoric pen and attaching individual identification labels to their housing crates. The animals were housed in the second animal breeding zone of Notus Co., Ltd. (Dar es salaam, Tanzania), with environmental conditions set at a temperature of 23 ± 3 °C, relative humidity of 55 ± 15%, ventilation frequency of 10–20 times/h, a 12 h light cycle (8 a.m. to 8 p.m.), and a brightness level of 150–300 Lux. Environmental parameters, including temperature, humidity, ventilation, and illumination in the animal room, were measured weekly. The animals were fed with experimental dog feed from Cargill Agrifrena Co., Ltd. (Seongnam, Republic of Korea), produced by Biopia (Stockholm, Sweden), with a daily intake of approximately 300 g, and water was freely accessible through an automatic watering system. During the acclimation, administration, and observation periods, the animals were individually housed in stainless-steel wire mesh cages (W 895 × L 795 × H 765 mm), and their cages were cleaned daily. After determining the health status of the animals during the acclimation period, their weights were measured and randomized to achieve a uniform distribution of mean weights within each group. 

### 4.2. Meniscectomy and Cranial Cruciate Ligament Resection (CCLx)

Beagle dogs underwent experimental surgery to mimic a specific disease, OA, without prior treatment, for research purposes. Before the surgery, the hair around the animals’ knee was shaved using clippers. The animals were anesthetized with Alfaxalone at a dosage of 3 mg/kg via intravenous administration for anesthesia induction, and maintenance was achieved using Isoflurane at a concentration of 0.5–3%, with the experiments conducted post-anesthesia. The surgical site was extensively disinfected with povidone and 70% alcohol. Subsequently, the skin of the right knee was incised, and blunt dissection was performed to expose the joint surface at the distal end of the femur. A cranial cruciate ligament resection created a defect on the medial articular surface. The wound was closed using 4-0 nylon sutures, and Maxon 2-0 suture (monofilament absorbable suture) was also used for suturing tissues below the skin. In the case of the left knee, only cranial cruciate ligament resection was performed. A week after meniscectomy, the animals underwent daily forced exercise (30 min/day) for 7 days. 

### 4.3. Sample Preparation and Evaluation of Drug Efficacy According to Dissolution Pattern

SKCPT tablets (SK Chemicals Co., Ltd., Seongnam, Republic of Korea), consisting of the exact composition of API in various concentrations and formulations, were formulated from *Clematis mandshurica*, *Trichosanthes kirilowii*, and *Prunella vulgaris* powdered extracts. SKCPT has been developed based on a previously licensed product (JOINS tablet [SKI306X, SK Chemicals Co., Ltd., Seongnam, Republic of Korea]) approved for the treatment of OA and rheumatoid arthritis in Korea by adjusting the usage and dose from 200 mg thrice daily to 300 mg twice daily. The diverse formulated tablets of SKI306X, including SKCPT, vehicle, and Celecoxib (Pfizer Inc., New York, NY, USA), were supplied by SK Chemicals. This study utilized refined extracts of SKI306X, which exhibit diverse release patterns, and composed of 30% ethanol extract (40→1). The impact of the release pattern on pharmacological efficacy was evaluated. The preparation of 200 mg and 300 mg IR, 300 mg MR, and 300 mg CR formulations involved the following steps: Initially, the binding solution was prepared by dissolving a binder in an appropriate solvent, which could be water, ethanol, isopropyl alcohol, or their mixtures. The extraction and colloidal silicon dioxide were then incorporated into the binding solution to form a wet granulation. The resulting wet granules were dried, blended with a filler, disintegrants, and then a lubricant and a glidant, and subsequently subjected to film coating using conventional techniques. The groups in the experiment were as follows: normal control (G1), vehicle control (G2), positive control (G3), 200 mg IR (G4), 300 mg IR (G5), 300 mg MR (G6, SKCPT), CR-1 (G7). Table 1 shows the detailed compositions. The prepared formulations underwent dissolution testing using the USP paddle method at 37.0 ± 0.5 °C. The dissolution test was conducted using a paddle speed of 50 rpm in 900 mL of water.

### 4.4. Sample Administration

All experimental groups started sample administration when the average gait assessment score reached ≥3. The samples were administered daily in the form of oral tablets. The animals were positioned in their natural state within their housing crates, their mouths were opened, and the tablet was placed on the inner side of the tongue. After closing their mouths, the posterior pharyngeal area was gently massaged to facilitate swallowing, and ingestion was confirmed. Approximately 10 mL of water was administered using a syringe. Table 2 shows the composition and administration details of each experimental group.

### 4.5. Measurement of Stifle Circumference

The rotational axis was determined by repeatedly extending and flexing the stifle joint. Using a ruler, the knee circumference was then measured at each time point at a location that was as consistent as feasible.

### 4.6. Gait Evaluation

Gait assessment was conducted pre-dosing and subsequently twice weekly post-dosing. Analysis of gait was performed in accordance with the evaluation criteria outlined in Appendix A below, and video footage was captured using a digital camera.

### 4.7. Incapacitance Test

In the case of the incapacitance test, a tester was utilized to measure the weight-bearing of both hind limbs, allowing for the analysis of weight distribution levels across the limbs. To confirm the effect of weight-bearing on a model of OA, hind limb weight-bearing was measured using an incapacitance tester (1029-S, Linton instrumentation, USA). In this experiment, beagle dogs induced with OA experienced more severe pain in the right hind limb, which underwent both cruciate ligament excision and medial meniscectomy, compared to the left hind limb, which only underwent cruciate ligament excision. Therefore, the dogs relied more on their left hind limb for support, leading to relatively lighter weight-bearing measurements on the right hind limb. During weight measurement, the weight of each hind paw was recorded separately, ensuring that the dog’s abdomen did not touch the sensor of the device. The weight-bearing ratio (%) was calculated using the formula provided below. The weight-bearing ratio is measured by pressing with the limb, and in normal cases, the weight distribution between both hind limbs is balanced, resulting in a weight-bearing ratio of 50% for each limb. However, as pain intensifies due to induced OA, the weight-bearing ratio of the affected hind limb decreases. Before and after administering the test substance, the weight distribution ratio of the hind limbs was measured using the incapacitance tester once a week for 8 weeks. The weight-bearing ratio formula is as follows:Weight-Bearing Ratio (%) = [Weight of OA Hind Limb/(Weight of Both Hind Limbs)] × 100

### 4.8. Histological and Histopathological Examination

After all the experiments were completed, euthanasia was carried out. Firstly, after confirming that the animal was in deep anesthesia using Isoflurane, 2 mEq/kg of Potassium Chloride solution was injected intravenously. On the day of necropsy, joint fluid was collected, and after euthanizing the animals, gross observations of the defect site were conducted, and photographs were taken. The defect site was excised and fixed in 10% neutral buffered formalin. The collected joint fluid was stored in an ultra-low temperature freezer at ≤−70 °C until analysis. Fixed tissue samples in 10% buffered formalin embedded in paraffin were sectioned into the slices. After undergoing routine tissue-processing procedures, including deparaffinization and dehydration, these slides were stained with hematoxylin & eosin (H&E) or Safranin-O dyeing reagents to compare histopathological evaluation among groups according to Appendix A. The immunohistochemical (IHC) staining process was initiated by deparaffinizing the slides using xylene, followed by rehydration via immersion in ethanol and subsequent hydration in water. After these preparatory steps, the inherent peroxidase activity was neutralized, and the slides’ permeabilization was facilitated. This was followed by a pre-blocking step involving the application of 10% normal goat serum for 1 h. An incubation period ensued, during which the slides were incubated overnight at 4 °C in the presence of each specific antibody. Each antibody was purchased as described below: the primary monoclonal MMP-13 antibody (R&D Systems, Minneapolis, MN, USA), Polyclonal TIMP1 antibody (MyBioSource, MBS2004304, San Diego, CA, USA), and Collagen II alpha 1 antibody (CiteAb, orb235107, Bath, UK). Post-incubation, the sections were cleansed and then exposed to horseradish peroxidase-conjugated secondary antibodies for an hour at ambient room temperature. The resultant enzymatic activity was made perceptible, and then, finally, the sections underwent counterstaining with H&E. After microscope observation, IHC staining was analyzed using an image analyzer (Zen 2.3 blue edition, Carl Zeiss, Oberkochen, Germany). H&E, Safranin-O, and IHC staining (MMP-13, TIMP-1, and collagen type II) were performed, followed by observation of histopathological changes using an optical microscope (Olympus BX53, Shinjuku, Japan) and camera (Olympus DP22, Shinjuku, Japan).

### 4.9. ELISA Analysis

Analysis of cytokines IL-1β, MMP-3, and TNF-α in the synovial fluid collected during post-mortem examination was conducted. The analysis was performed using the following ELISA kits: Canine IL-1β ELISA Kit (Abcam, Ab273170, Waltham, MA, USA), Canine MMP-3 ELISA Kit (Cusabio, CSB-E13621c, Houston, TX, USA), and Canine TNF-α ELISA Kit (R&D system, CATA00, Minneapolis, MN, USA).

### 4.10. Statistical Analysis

Both parametric multiple comparison procedures and non-parametric multiple comparison procedures were utilized to analyze the results. The normality of the data was assessed using the Shapiro–Wilk test. In cases where the data exhibited normality, one-way ANOVA was performed for parametric multiple comparisons. For significance, Dunnett’s multiple comparison test was utilized to analyze the significant differences among the test groups. When the data did not follow a normal distribution, non-parametric multiple comparison analysis was performed using the Kruskal–Wallis H-test. If the result was significant, post hoc analysis using the Mann–Whitney U test was conducted to identify significant differences among the experimental groups. Statistical analysis was performed using Prism 7.04 (GraphPad Software Inc., San Diego, CA, USA), and a *p*-value of less than 0.05 was considered statistically significant.

## 5. Conclusions

In this study, using beagle dogs as subjects, the therapeutic efficacy of SKI306X on OA was investigated using many evaluations, with the group treated with SKCPT demonstrating the most promising results, thereby indicating potential advantages in the treatment of arthritis. SKCPT has a specific release pattern, resulting in excellent arthritis control even when administered twice daily.

## Figures and Tables

**Figure 1 ijms-24-14972-f001:**
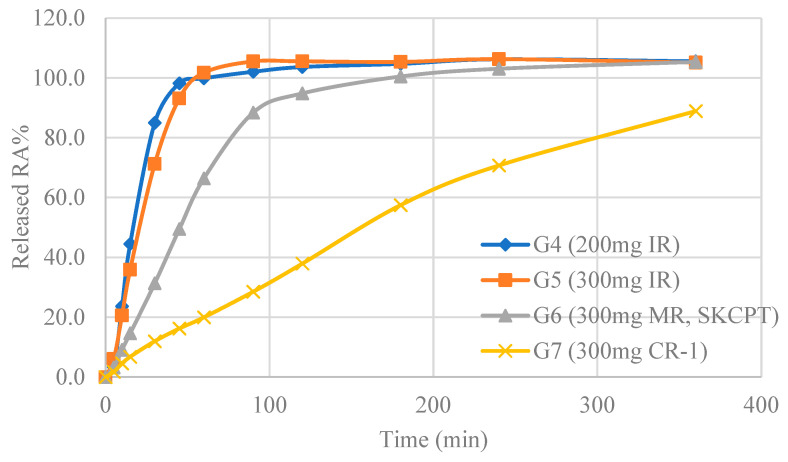
The results of dissolution test of various formulation types made with SKI306Xs. The dissolution test was conducted using the paddle method with a stirring speed of 50 rpm and performed at a dissolution temperature of 37.0 ± 0.5 °C. Dissolution was carried out in a 900 mL solution in purified water for each formulation. To improve sample stability, acetonitrile pre-treatment was employed before analysis, and the dissolution test for ‘Rosmarinic acid’ was verified.

**Figure 2 ijms-24-14972-f002:**
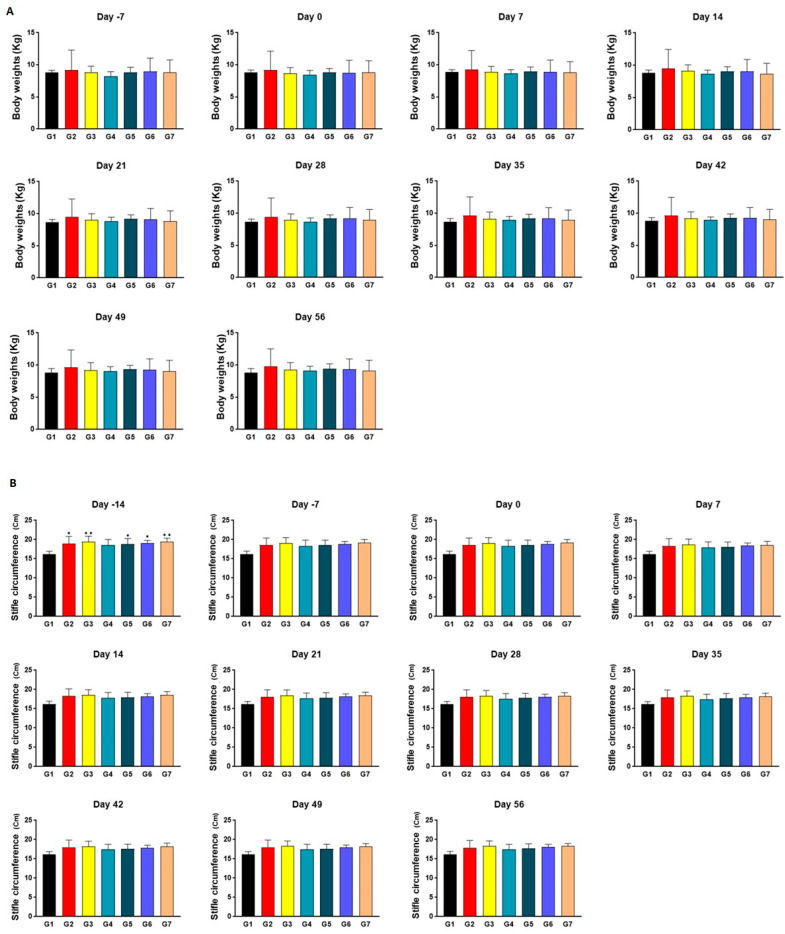
The results of clinical changes in all the experimental groups. (**A**) The weight of each individual was measured weekly, both one week prior to the administration of the test substance and throughout the following eight weeks. (**B**) The measurement of stifle circumference was carried out weekly, starting 14 days prior to the initiation of substance administration and continuing for a duration of 8 weeks. This was conducted to assess the extent of joint swelling. Data are expressed as mean ± S.D. Day of first test article administration was designated day 0. G1: normal control, G2: vehicle control, G3: positive control, G4: 200 mg IR, G5: 300 mg IR, G6: 300 mg MR, G7: CR-1. **/* A significant difference at *p* < 0.01/*p* < 0.05 level compared to the G1.

**Figure 3 ijms-24-14972-f003:**
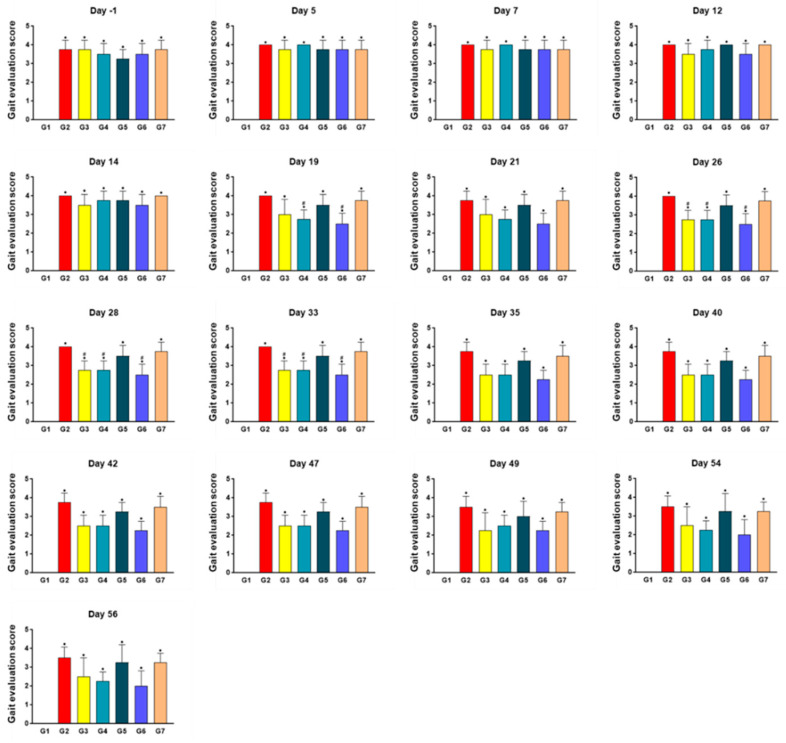
The result of gait evaluation score in a canine model of OA. As a kinetics assessment, gait evaluation was recorded twice a week and conducted using a pressure-sensing walkway. For each dog at each time point, at least three acceptable passes (consisting of 3–5 gait cycles) were recorded, along with video documentation. Data are expressed as mean ± S.D. Day of first test article administration was designated day 0. G1: normal control, G2: vehicle control, G3: positive control, G4: 200 mg IR, G5: 300 mg IR, G6: 300 mg MR, G7: CR-1. * A significant difference at *p* < 0.05 level compared to the G1. # A significant difference at *p* < 0.05 level compared to the G2.

**Figure 4 ijms-24-14972-f004:**
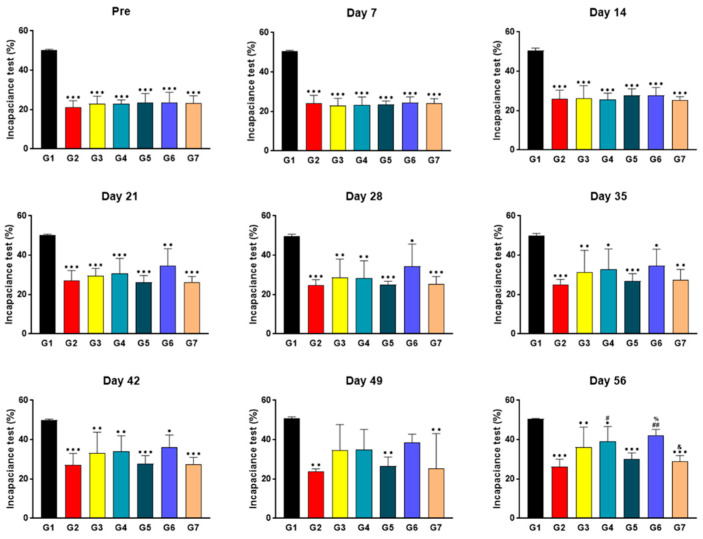
The results of incapacitance test in a canine model of OA. To measure the fluid accumulation or edema, an incapacitance test was performed by applying pressure to the limb and assessing changes in capacitance or volume. The post-weight-bearing distribution was measured using an incapacitance tester once a week for 8 weeks, both before and after administration. Data are expressed as mean ± S.D. Day of first test article administration was designated day 0. G1: normal control, G2: vehicle control, G3: positive control, G4: 200 mg IR, G5: 300 mg IR, G6: 300 mg MR, G7: CR-1. ***/**/* A significant difference at *p* < 0.001/*p* < 0.01/*p* < 0.05 level compared to the G1. ##/# A significant difference at *p* < 0.01/*p* < 0.05 level compared to the G2. % A significant difference at *p* < 0.05 level compared to the G5. & A significant difference at *p* < 0.05 level compared to the G6.

**Figure 5 ijms-24-14972-f005:**
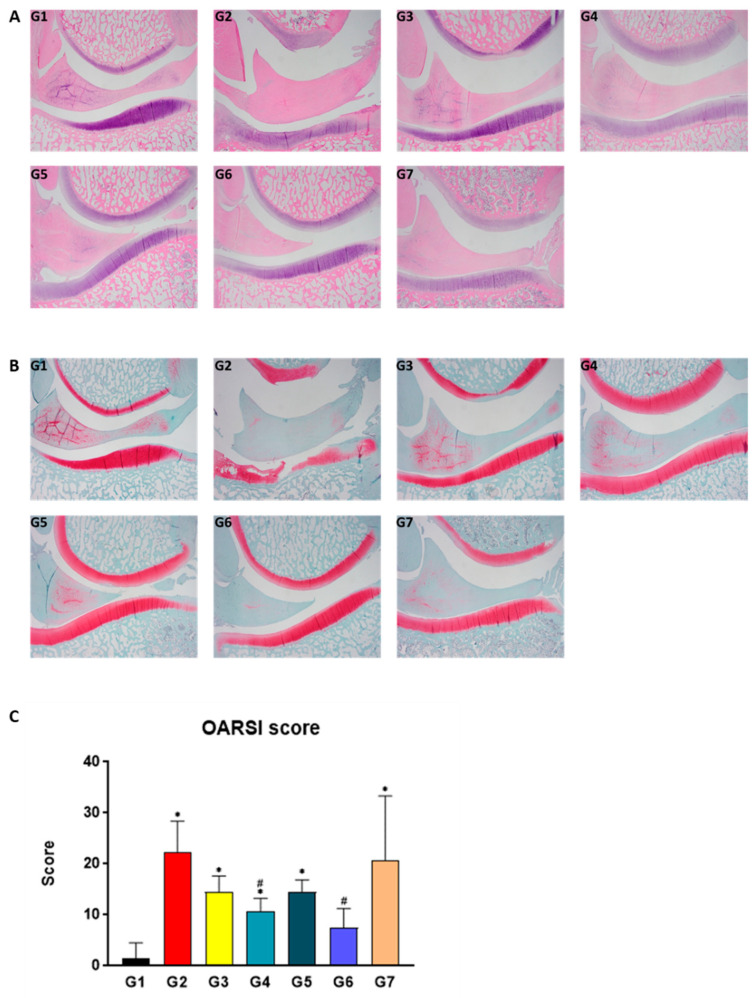
The results of histological examination in a canine model of OA. Quantification of cartilage structure, chondrocyte pathology, proteoglycan staining, lining cell characteristics, cell infiltration characteristics, and hyperplasia characteristics was performed based on the results of (**A**) H&E and (**B**) Safranin-O staining. These parameters were graded according to their severity, and compared among groups, with the results expressed as (**C**) the OARSI score. Data are expressed as mean ± S.D. Day of first test article administration was designated day 0. G1: normal control, G2: vehicle control, G3: positive control, G4: 200 mg IR, G5: 300 mg IR, G6: 300 mg MR, G7: CR-1. * A significant difference at *p* < 0.01/*p* < 0.05 level compared to the G1. # A significant difference at *p* < 0.05 level compared to the G2.

**Figure 6 ijms-24-14972-f006:**
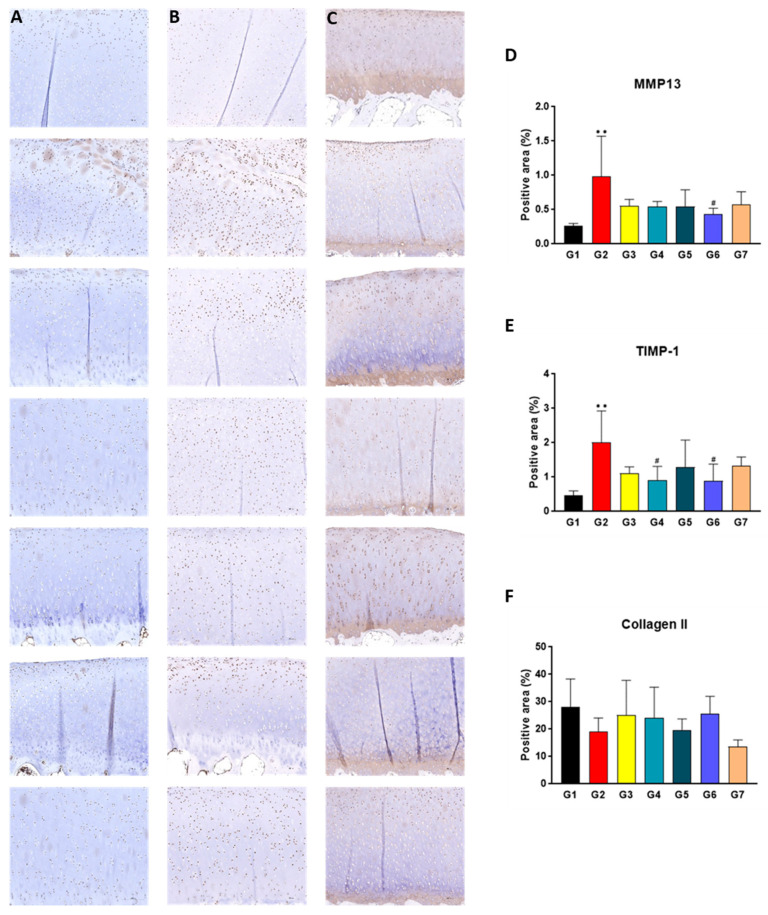
The results of histopathological examination in a canine model of OA. The paraffin-embedded tissue slides were stained with each primary antibody, (**A**) MMP-13, (**B**) TIMP-1, and (**C**) collagen type II. (**D**–**F**) As result of IHC staining, percentage of positive staining area per total skin area was calculated using Image J. Data are expressed as mean ± S.D. Day of first test article administration was designated day 0. G1: normal control, G2: vehicle control, G3: positive control, G4: 200 mg IR, G5: 300 mg IR, G6: 300 mg MR, G7: CR-1. ** A significant difference at *p* < 0.01 level compared to the G1. # A significant difference at *p* < 0.05 level compared to the G2.

**Figure 7 ijms-24-14972-f007:**
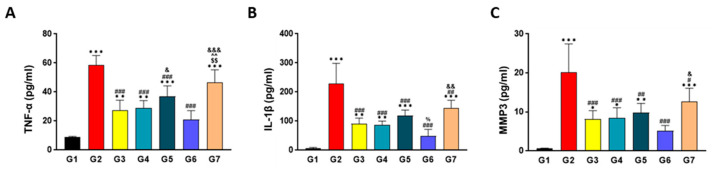
The results of cytokines analysis in a canine model of OA. Each cytokine analysis of (**A**) TNF-α, (**B**) IL-1β, and (**C**) MMP-3 was carried out on synovial fluid obtained during post-mortem examination, utilizing specific ELISA kits for each cytokine. Data are expressed as mean ± S.D. Day of first test article administration was designated day 0. G1: normal control, G2: vehicle control, G3: positive control, G4: 200 mg IR, G5: 300 mg IR, G6: 300 mg MR, G7: CR-1. ***/**/* A significant difference at *p* < 0.001/*p* < 0.01/*p* < 0.05 level compared to the G1. ###/##/# A significant difference at *p* < 0.001/*p* < 0.01/*p* < 0.05 level compared to the G2. $$ A significant difference at *p <* 0.01 level compared to the G3. ^^ A significant difference at *p <* 0.01 level compared to the G4. % A significant difference at *p* < 0.05 level compared to the G5. &&&/&&/& A significant difference at *p* < 0.001/*p* < 0.01/*p* < 0.05 level compared to the G6.

**Table 1 ijms-24-14972-t001:** The manufacturing information of tablets for dissolution assessment.

	G4	G5	G6	G7
SKI306X	200	300	300	300
Hydrophobic collodial silica (Aerosil R972)	10	15	20	20
Corn starch	50			
Povidon K30		5		
Ethylcellulose			4	4
Microcrystalline cellulose	113	85	115	115
Kollidon SR				50
Sodium starch glycolate	25			
Crospovidone		15		
Croscarmellose sodium		25	10	10
Magnesium stearate	2	5		
Sodium stearyl fumarate			5	5
Opadry	30	27	29	30
Total weight	430	477	483	534

**Table 2 ijms-24-14972-t002:** The composition and administration details of each experimental group.

ID	Sex	Number of Animals	Animal Number	Meniscectomy	Administrated Substances	Number of Administrations	Dosage (mg)	Dosage (Tablet)
G1	M	4	1–4	N				
G2	M	4	5–8	Y	Vehicle	b.i.d	N/A	1T
G3	M	4	9–12	Y	Celecoxib	q.d.	200	1T
G4	M	4	13–16	Y	IR	t.i.d	200	1T
G5	M	4	17–20	Y	IR	b.i.d	300	1T
G6	M	4	21–24	Y	MR	b.i.d	300	1T
G7	M	4	25–28	Y	CR-1	b.i.d	300	1T

## Data Availability

The data presented in this study are available on request from the corresponding author.

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
