# Peer review of "Effects of SKCPT on Osteoarthritis in Beagle Meniscectomy and Cranial Cruciate Ligament Transection Models"

_ijms, 2023, doi:10.3390/ijms241914972_

Round 1
Reviewer 1 Report
The importance of the topic discussed is underlined. The work is well written and follows an in vivo experimental project.. The experimental groups are well divided and identifiable as are the results obtained which open up further experimental studies. The statistical investigation is good and the bibliographical references are adequate.
Author Response
We appreciate reviewers’ valuable comments and the opportunity to re-submit the manuscript, entitled “Effects of SKCPT on Osteoarthritis in Beagle Meniscectomy and Cranial Cruciate Ligament Transection Models”. Reviewers’ comments are insightful and we have revised the manuscript accordingly.
*The title has been changed due to a change in terminology at the reviewer's request.
Thank you for your reference.
Thank you for your all insightful comments.
Reviewer 2 Report
Overall the paper is well written and organized. The abstract is lacking in details about the study design (like sample size) and reads more like a review. That can be adjusted easily. Acronyms should be defined or explained when they are first used throughout the abstract and introduction. The experimental design seems appropriate. I had a very difficult time reading and interpreting the figures based on their size and how they are laid out in this version of the manuscript. I would recommend the authors revisit these to maximize clarity and readability to the reader. There is little reason to create the figures if they cannot be read or convey the information they are meant to. The narrative about the results is light on content and refers the authors to the figures which are difficult to decipher so I cannot comment on the quality of the results until these are revised. The discussion is well organized and the methods are clear. The authors do not overstate their results but could include more regarding the limitations of this study. Overall with a few revisions and revisiting the figures this manuscript is a good candidate for future publication.
Author Response
We appreciate reviewers’ valuable comments and the opportunity to re-submit the manuscript, entitled “Effects of SKCPT on Osteoarthritis in Beagle Meniscectomy and Cranial Cruciate Ligament Transection Models”. Reviewers’ comments are insightful and we have revised the manuscript accordingly.
*All author names and their order remain the same; however, there was an error in the author information provided to the journal. We kindly request a correction. The first author of the paper is solely Hye Min Kim, and we kindly request the correction once again. Apart from this fact, all other author details remain unchanged, and authorship remains the same.
*The modified part is marked in red in the 'ijms-2610633_Revised manuscript'.
*The title has been changed due to a change in terminology at the reviewer's request.
Thank you for your reference.
Response: Thank you for your all insightful comments.
We have made some modifications to further clarify the descriptions. The formatting adheres to the guidelines specified by the journal. We hope that these changes address the readability issues you mentioned.
Regarding SKI306X, which can be considered the starting point of this research, there is a substantial body of existing research, and this paper elaborates on the efficacies discovered. Furthermore, the authors have made efforts to create a formulation that can maximize these efficacies. Beyond the limitations of this study, we are committed to conducting further research to ultimately offer the maximum efficacy to humans.
Additionally, if there are any specific matters you would like us to address or suggestions you would like to provide, please let us know, and we will make every effort to accommodate them.
→ Modified section: Page 3
→ Revised manuscript: As part of this, the meniscectomy and cranial cruciate ligament transection (CCLx)-treated Beagle dog was used in this experiment. The cranial cruciate ligament (CCL) in dogs pri-marily serves as the principal restraint against anterior tibial displacement relative to the femur and safeguards against stifle joint hyperextension; its insufficiency results in modified joint biomechanics, heightened femorotibial cartilage stresses, and the rapid onset and progression of osteoarthritis, along with an elevated vulnerability to secondary meniscal injuries.

Reviewer 3 Report
The study concerns osteoarthritis, which is indeed a serious problem. In the introduction, the authors examine the issue of this condition in detail. Beagle dogs were used for the study. This research model is objectionable and the authors should justify the choice of this animal species more strongly. Such a study could have been carried out on eg. miniature pigs. In dogs, osteoarthritis of the knee joint is marginal and, if present, is associated with rupture or tear of the cranial cruciate ligament. The research methodology is appropriate, but the description of the procedure applied to the animals is too laconic. The authors should use the canine-specific anatomicvzz mainstreaming of the structures of the knee joint.
Discussion is appropriate. The literature is sufficient.
4. Please state the method of animal euthanasia
2.5. Please specify which microscope and camera were used
4. Please indicate the age of the dogs used in the study.
4. 2. In dogs, as in other quadrupedal animals, the human equivalent of the anterior cruciate ligament is the cranial cucial ligament - please use the appropriate nomenclature and possibly explain to the reader in two words (https://www.wava-amav.org/wava-documents.html)
4.2 4.2 Were soluble subcutaneous sutures used, as I understand Nylon 4-0 was used for the skin anastomosis?
4.2. Please describe the anaesthetic procedure
Author Response
We appreciate reviewers’ valuable comments and the opportunity to re-submit the manuscript, entitled “Effects of SKCPT on Osteoarthritis in Beagle Meniscectomy and Cranial Cruciate Ligament Transection Models”. Reviewers’ comments are insightful and we have revised the manuscript accordingly.
*All author names and their order remain the same; however, there was an error in the author information provided to the journal. We kindly request a correction. The first author of the paper is solely Hye Min Kim, and we kindly request the correction once again. Apart from this fact, all other author details remain unchanged, and authorship remains the same.
*The modified part is marked in red in the 'ijms-2610633_Revised manuscript'.
*The title has been changed due to a change in terminology at the reviewer's request.
Thank you for your reference.
Response: Thank you for your all insightful comments.
The reason for using Beagle dogs as a model in this study is as mentioned in the 'Introduction', primarily because the etiology of arthritis is similar between humans and dogs. According to reference studies, dogs have shown a remarkable similarity to humans in terms of osteoporosis-related aspects among large animals, including dogs, goats, sheep, and horses. In the field of osteoporosis research, this model is one of those recommended by the 'Ministry of Food and Drug Safety' for non-clinical studies. Furthermore, it is the most feasible model available, which is why we conducted our experiments using this particular model.
→ Mentioned section: Page 2-3
→ Relevant research papers: Models of Osteoarthritis: Relevance and New Insights (DOI: 10.1007/s00223-020-00670-x)/A review of translational animal models for knee osteoarthritis (DOI: 10.1155/2012/764621)
4. Please state the method of animal euthanasia
Response: Thank you for your insightful comment. After all the experiments were completed, euthanasia was carried out. First, after confirming that the animal was in deep anesthesia using Isoflurane, 2 mEq/kg of Potassium Chloride solution was injected intravenously. Since this information is about euthanasia for joint fluid and tissue collection, it was added to the first sentence of '4.7. Histological and histopathological examination' in Materials and Methods.
→ Modified section: Page 14
→ Revised manuscript: After all the experiments were completed, euthanasia was carried out. First, after con-firming that the animal was in deep anesthesia using Isoflurane, 2 mEq/kg of Potassium Chloride solution was injected intravenously.
2.5. Please specify which microscope and camera were used
Response: Thank you for your pointing out it.
The requested information has been provided in detail as follows:
Microscope: Olympus BX53, Japan
Camera: Olympus DP22, Japan
→ Modified section: Page 14
→ Revised manuscript: ~and camera (Olympus DP22, Japan).
4. Please indicate the age of the dogs used in the study.
Response: Thank you for your critical comment. The age of the Beagle models used in this experiment ranged from 12 to 17 months.
→ Modified section: Page 11
→ Revised manuscript: ~(12-17 months)
4. 2. In dogs, as in other quadrupedal animals, the human equivalent of the anterior cruciate ligament is the cranial cucial ligament - please use the appropriate nomenclature and possibly explain to the reader in two words (https://www.wava-amav.org/wava-documents.html)
Response: We do appreciate your significant comment. Overall, the terminology has been revised. As suggested, we have used the more accurate term "Cranial cruciate ligament" instead of "Anterior cruciate ligament" and have incorporated this change into the text.
→ Modified section: Page 1 and 11
→ Revised manuscript: '~Cranial cruciate ligament resection' or 'CCLx'
4. 2. Were soluble subcutaneous sutures used, as I understand Nylon 4-0 was used for the skin anastomosis?
Response: We do appreciate your valuable comment. For suturing tissues below the skin but outside the subcutaneous layer, Maxon 2-0 sutures (monofilament absorbable sutures) were employed. As per your request for more detail, the information has been elaborated upon.
→ Modified section: Page 11
→ Revised manuscript: The wound was closed using 4-0 nylon sutures, and Maxon 2-0 suture (monofilament ab-sorbable suture) was also used for suturing tissues below the skin.
4.2. Please describe the anaesthetic procedure
Response: Thank you for your critical comment. As per your request for additional detail, anesthesia induction was carried out with Alfaxalone at a dosage of 3 mg/kg via intravenous administration. For maintenance, Isoflurane at a concentration of 0.5-3% was utilized, and the experiments were conducted post-anesthesia.
→ Modified section: Page 11
→ Revised manuscript: The animals were anesthetized with Alfaxalone at a dosage of 3 mg/kg via intravenous administration for anesthesia induction, and maintenance was achieved using Isoflurane at a concentration of 0.5-3%, with the experiments conducted post-anesthesia.

Reviewer 4 Report
Dear authors,
Your work has relevant information. In my opinion, its writing did not keep up with this relevance and an effort has to be made to rewrite the work to enhance it.
Abstract
The authors defined osteoarthritis but did not ACLT -
Introduction
"In 2021, >22% of individuals aged ≥40 years were diagnosed with knee OA" Where?
"This medication was administered as the positive control group in this experiment to compare pain relief and anti-inflammatory abilities in joint diseases. Please change this sentence to MM section
Please refer specific Coxibs side effects.
I think this sentence is the main objective and should appear at the end of the introduction.
"In this study, the efficacy of the SKCPT tablet was examined to minimize the shortcomings of SKI306X and maximize its advantages."
SKCPT active principle instead tablet,...
gastrointestinal irritation ????
Repeated ideas,...
"However, considering the significant anatomical and pathological differences among species, histological examination remains the benchmark for outcome evaluation in canine OA models [20, 21].
Besides, given the significant anatomic and pathologic disparities among species histological examination remains the authoritative standard for evaluating outcomes in canine OA models [18]."
Results
The results must be presented in accordance with the methods statement in the respective section.
Legend of figure 2 - "After acquiring a total of 28 beagle dogs, with four dogs per group, a 10-day adaptation period was provided before randomly dividing them into groups." This information should only be allocated to MM section.
statistic p in italics
"Through histological examination, the total cartilage score, which quantifies the degree of cartilage damage, and the total synovial score, which evaluates the structure of the synovial tissue along with the escalation of inflammatory cells, vascular dilation, and tissue swelling, were assessed via hematoxylin and eosin (H&E)(Figure 5A and Figure S1) and Safranin O staining (Figure 5B). Additionally, the OARSI score was utilized to evaluate the severity of OA through the aforementioned staining methods" MM section
The histological figures must contain a scale bar and the observed parameters that are different between the various groups must be marked on them.
Materials and Methods
4.1.
KNOTUS Co., Ltd. (Incheon, Korea) [Approval number: KNOTUS Institutional Animal Care and Use Committee (IACUC)]. The number is not provided
What is the KNOTUS Co., Ltd. (Incheon, Korea) ? This information does not appear in the affiliation of any author.
The animals were housed in the second animal breeding zone of KNOTUS Co.,Ltd.
The animals were not distributed into groups.
4.2.
The animals were anesthetized,
Please describe the anesthetic protocol.
The description of the surgical procedure is not clear.
4.4. Sample Administration
What is the average gait assessment score. Please provide a reference.
4.7. How was euthanasia performed?
The authors must explain how they controlled the human endpoints while carrying out this work.
Discussion
In my opinion, the discussion section should be rewritten as it is limited to describing the results and there is no substantial comparison with results from similar works by other authors.
"drug delivery systems, namely immediate release (IR), modified release (MR), and controlled release (CR)."
This information is not included in the Material and methods section, which should be rewritten with more attention to all these details.
"The groups in the experiment were as follows: normal control (G1), vehicle control (G2), positive control (G3), 200 mg IR (G4), 300 mg IR (G5), 300 mg MR (G6, SKCPT), CR-1 (G7)." Please rewrite this information in MM section.
"Joint swelling was higher in the test group than in the normal group" Which of the experimental groups are you referring to?
Where can you find the description of the "OARSI score" in the MM section?
Author Response
We appreciate reviewers’ valuable comments and the opportunity to re-submit the manuscript, entitled “Effects of SKCPT on Osteoarthritis in Beagle Meniscectomy and Cranial Cruciate Ligament Transection Models”. Reviewers’ comments are insightful and we have revised the manuscript accordingly.
*All author names and their order remain the same; however, there was an error in the author information provided to the journal. We kindly request a correction. The first author of the paper is solely Hye Min Kim, and we kindly request the correction once again. Apart from this fact, all other author details remain unchanged, and authorship remains the same.
*The modified part is marked in red in the 'ijms-2610633_Revised manuscript'.
*The title has been changed due to a change in terminology at the reviewer's request.
Thank you for your reference.
Abstract
The authors defined osteoarthritis but did not ACLT -
Response: Thank you for your insightful comment. Overall, the terminology has been revised to further clarify the descriptions. We have used the more accurate term "Cranial cruciate ligament" instead of "Anterior cruciate ligament" and have incorporated this change into the text. And then, as you suggested, we defined 'CCLx' in the introduction.
→ Modified section: Page 3
→ Revised manuscript: As part of this, the meniscectomy and cranial cruciate ligament transection (CCLx)-treated Beagle dog was used in this experiment. The cranial cruciate ligament (CCL) in dogs primarily serves as the principal restraint against anterior tibial displacement relative to the femur and safeguards against stifle joint hyperextension; its insufficiency results in modi-fied joint biomechanics, heightened femorotibial cartilage stresses, and the rapid onset and progression of osteoarthritis, along with an elevated vulnerability to secondary meniscal injuries.
Introduction
"In 2021, >22% of individuals aged ≥40 years were diagnosed with knee OA" Where?
Response: Thank you for your pointing out it. As in the reference section, it is mentioned in the paper reference number [2].
→ Mentioned reference: Yao, Q.; Wu, X.; Tao, C.; Gong, W.; Chen, M.; Qu, M.; Zhong, Y.; He, T.; Chen, S.; Xiao, G., Osteoarthritis: pathogenic signaling pathways and therapeutic targets. Signal transduction and targeted therapy 2023, 8, (1), 56.
"This medication was administered as the positive control group in this experiment to compare pain relief and anti-inflammatory abilities in joint diseases. Please change this sentence to MM section
Response: Thank you for your critical comment. With reference to what you said, we removed this sentence because it seemed to fit the flow and location 'Introduction'.
Please refer specific Coxibs side effects.
Response: We do appreciate your significant comment. As you instructed, we refered it in detail.
→ Modified section: Page 2
→ Revised manuscript: When using Celecoxib to treat OA, potential side effects include gastrointestinal problems, cardiovascular risks, edema, kidney issues, allergic reactions, liver function impact, gas-trointestinal bleeding, headache, dizziness, and an increased risk of blood clots [9, 10].
I think this sentence is the main objective and should appear at the end of the introduction.
"In this study, the efficacy of the SKCPT tablet was examined to minimize the shortcomings of SKI306X and maximize its advantages."
Response: Thank you for your valuable comments. As you said, the location has been changed to the last part of the introduction.
→ Modified section: Page 3
gastrointestinal irritation ????
Response: Thank you for your valuable question. 'Gastrointestinal irritation' refers to the irritation or inflammation of the lining of the gastrointestinal tract, which includes the stomach, small intestine, and large intestine. Gastrointestinal irritation manifests as nausea, vomiting, diarrhea, and abdominal pain. It is the symptom of gastrointestinal disorder, and we are attaching a link to the journal related to the use of this term.
→ Relative journal site: https://www.sciencedirect.com/topics/pharmacology-toxicology-and-pharmaceutical-science/gastrointestinal-irritation
"However, considering the significant anatomical and pathological differences among species, histological examination remains the benchmark for outcome evaluation in canine OA models [20, 21].
Besides, given the significant anatomic and pathologic disparities among species histological examination remains the authoritative standard for evaluating outcomes in canine OA models [18]."
Response: Thanks for the good point. Repetitive content has been revised into one.
→ Modified section: Page 2
Results
The results must be presented in accordance with the methods statement in the respective section.
Legend of figure 2 - "After acquiring a total of 28 beagle dogs, with four dogs per group, a 10-day adaptation period was provided before randomly dividing them into groups." This information should only be allocated to MM section.
Response: Thank you for your pointing out it. The part you mentioned has been deleted from ‘Legend of figure 2’, and the same content can be found in '4.1. Animals' of the MM section.
→ Modified section: Page 4
→ Mentioned section: Page 11
statistic p in italics
Response: Thank you for your critical comment. As you mentioned, we changed every 'p' to italics.
"Through histological examination, the total cartilage score, which quantifies the degree of cartilage damage, and the total synovial score, which evaluates the structure of the synovial tissue along with the escalation of inflammatory cells, vascular dilation, and tissue swelling, were assessed via hematoxylin and eosin (H&E)(Figure 5A and Figure S1) and Safranin O staining (Figure 5B). Additionally, the OARSI score was utilized to evaluate the severity of OA through the aforementioned staining methods" MM section
The histological figures must contain a scale bar and the observed parameters that are different between the various groups must be marked on them.
Response: We do appreciate your valuable comment. Due to screen quality reasons, it was difficult to submit images including scale bars. We sincerely ask for your understanding. In the case of the parameters you mentioned, the standards are provided in the supplementary Tables.
→ Mentioned section: Page 14
→ ~, these slides were stained with Hematoxylin & Eosin (H&E) or Safranin-O dyeing reagents to compare histopathological evaluation among groups according to Table S2 to Table S7 provided.
Materials and Methods
4.1.
KNOTUS Co., Ltd. (Incheon, Korea) [Approval number: KNOTUS Institutional Animal Care and Use Committee (IACUC)]. The number is not provided
What is the KNOTUS Co., Ltd. (Incheon, Korea) ? This information does not appear in the affiliation of any author.
Response: Thank you for your critical and detail comment. KNOTUS is a non-clinical Contract Research Organization (CRO) affiliated with 'HLB Biosciences Inc.,' where experimental equipment has been established for research purposes. It has served as the location for this experiment and has conducted a wide range of animal experiments as evidenced by numerous publications. Attached are relevant research papers.
→ Modified section: Page 11
→ Revised manuscript: All animal experiments were cwas approved by the Institutional Animal Care and Use Committee of Knotus (No. IACUC 21-KE-199).
→ Relevant research papers: Anti-Obesity Effect of Standardized Extract of Microalga Phaeodactylum tricornutum Containing Fucoxanthin (doi: 10.3390/md17050311)/Reproducible Large-Scale Isolation of Exosomes from Adipose Tissue-Derived Mesenchymal Stem/Stromal Cells and Their Application in Acute Kidney Injury (doi: 10.3390/ijms21134774)/Preventive effect of biodegradable stents on biliary stricture and fibrosis after biliary anastomosis in a porcine model (doi: 10.4174/astr.2022.102.2.90.)/Development of a gastroretentive delivery system for acyclovir by 3D printing technology and its in vivo pharmacokinetic evaluation in Beagle dogs(doi: 10.1371/journal.pone.0216875)
4.2.
The animals were anesthetized,
Please describe the anesthetic protocol.
The description of the surgical procedure is not clear.
Response: Thank you for your critical comment. As per your request for additional detail, anesthesia induction was carried out with Alfaxalone at a dosage of 3 mg/kg via intravenous administration. For maintenance, Isoflurane at a concentration of 0.5-3% was utilized, and the experiments were conducted post-anesthesia.
→ Modified section: Page 11
→ Revised manuscript: The animals were anesthetized with Alfaxalone at a dosage of 3 mg/kg via intravenous administration for anesthesia induction, and maintenance was achieved using Isoflurane at a concentration of 0.5-3%, with the experiments conducted post-anesthesia.
4.4. Sample Administration
What is the average gait assessment score. Please provide a reference.
Response: Thank you for your comment. The values in Figure 3 represent the average gait assessment score.
4.7. How was euthanasia performed?
The authors must explain how they controlled the human endpoints while carrying out this work.
Response: Thank you for your insightful comment. After all the experiments were completed, euthanasia was carried out. First, after confirming that the animal was in deep anesthesia using Isoflurane, 2 mEq/kg of Potassium Chloride solution was injected intravenously. Since this information is about euthanasia for joint fluid and tissue collection, it was added to the first sentence of '4.7. Histological and histopathological examination' in Materials and Methods.
→ Modified section: Page 14
→ Revised manuscript: After all the experiments were completed, euthanasia was carried out. First, after con-firming that the animal was in deep anesthesia using Isoflurane, 2 mEq/kg of Potassium Chloride solution was injected intravenously.
Discussion
In my opinion, the discussion section should be rewritten as it is limited to describing the results and there is no substantial comparison with results from similar works by other authors.
Response: Thank you for your valuable comment. In this paper, we primarily focus on comparing the formulation and dosage of the existing marketed drug, SKI306, with a particular emphasis in the Introduction section on the efficacy of SKI306 as revealed in previous research, highlighting the necessity of our study. The results of these experiments confirm that SKCPT formulation exhibits the most effective efficacy, advancing upon the findings of previous papers and presenting this paper as a distinctive contribution. If there are any additional points that need to be addressed or included, please provide specific details, and we will do our best to follow your comment accordingly.
"drug delivery systems, namely immediate release (IR), modified release (MR), and controlled release (CR)."
This information is not included in the Material and methods section, which should be rewritten with more attention to all these details.
Response: Thank you for your thoughtful advice. Due to the structure of this journal, the Material and Methods section is placed after the Discussion section. To maintain the smooth flow of the paper, we have incorporated the content into the Discussion section and, in order to avoid redundancy, it has been omitted from the MM section. Your understanding is greatly appreciated.
"The groups in the experiment were as follows: normal control (G1), vehicle control (G2), positive control (G3), 200 mg IR (G4), 300 mg IR (G5), 300 mg MR (G6, SKCPT), CR-1 (G7)." Please rewrite this information in MM section.
Response: Thank you for your advice. As you suggested, we have mentioned it once again in the MM section under '4.3. Sample preparation and evaluation of drug efficacy according to dissolution pattern.'
→ Modified section: Page 12
"Joint swelling was higher in the test group than in the normal group" Which of the experimental groups are you referring to?
Response: Thank you for your detailed comment. When you review the results in Figures 2 and 5, you will observe an increase in the experimental group compared to the normal control (G1).
Where can you find the description of the "OARSI score" in the MM section?
Response: Thank you for your insightful comment. The criteria for the OARSI score are based on the results obtained from staining, and the staining method is referenced in the '4.7. Histological and histopathological examination' of the MM section. Additionally, the criteria for the OARSI score are mentioned in the supplementary tables of the MM section.
→ Mentioned section: Page 14
→ ~, these slides were stained with Hematoxylin & Eosin (H&E) or Safranin-O dyeing reagents to compare histopathological evaluation among groups according to Table S2 to Table S7 provided.

Round 2
Reviewer 3 Report
The authors have made corrections that significantly improve its reception. They have modified the nomenclature and described the procedures used.
Author Response
We sincerely appreciate your all comments of the revision. Thanks to your feedback, we have had the opportunity to enhance our paper and make it better.
Reviewer 4 Report
Dear authors,
There remain some questions that have not been adequately clarified.
Abstract
1 - The meniscectomy and cranial cruciate ligament transection treated Beagle dog model was used to investigate the efficacy of modified release SKI306X (SKCPT) formulation from Clematis mandshurica, Prunella vulgaris, and Trichosanthes kirilowii in managing arthritis. (the same at introduction section)
2 - Repeated ideas
The group treated with 300 mg of Modified Release (MR) formulation demonstrated superior efficacy. Pro-inflammatory cytokines and matrix metalloproteinases were significantly lower in the SKCPT-treated groups than in the control group. This research found that the SKCPT reduces arthritic lesions and improves abnormal gait. The 300 mg MR formulation was more efficacious than others, suggesting a promising approach for managing OA symptoms and addressing disease pathogenesis.
Introduction
In the article "Quicke, J. G., Conaghan, P. G., Corp, N. & Peat, G. Osteoarthritis year in review 2021: epidemiology & therapy. Osteoarthr. Cartil. 30, 196–206 (2022)." I can't find any reference to the phrase: "In 2021, >22% of adults older than had knee OA, and it is estimated that over 500 million individuals are currently affected by OA worldwide.", cited by the article "Yao, Q.; Wu, X.; Tao, C.; Gong, W.; Chen, M.; Qu, M.; Zhong, Y.; He, T.; Chen, S.; Xiao, G., Osteoarthritis: pathogenic signaling pathways and therapeutic targets. Signal transduction and targeted therapy 2023, 8, (1), 56." which you in turn mentioned. I recommend looking for a new reference and clarifying which population (space-time) the epidemiological reference refers to.
Materials and Methods
4.3
Clematis mandshurica, Prunella vulgaris, and Trichosanthes kirilowii, powdered extracts. as in the abstract
Table 1. Magnesium stearate
Results
"The paraffin-embedded tissue slides were stained with each dye after deparaffinization and rehydration processes. Quantification of cartilage structure, chondrocyte pathology, proteoglycan staining, lining cell characteristics, cell infiltration characteristics, and hyperplasia characteristics was performed based on the results of (A) H&E and (B) Safranin-O staining. These parameters were graded according to their severity, and compared among groups, with the results expressed as (C) the OARSI score."
This information is provided in the Material and Methods section and is not expected in a figure legend. The captions should take the opportunity to describe the differences found in the various slides, which clearly do not appear in the captions of this work. Once again, scale bars are requested to be placed on photographs of all histological preparations.
the same for Figure 6 "After deparaffinization, rehydration, antigen retrieval, and blocking processes, the paraffin-embedded tissue slides were stained with each primary antibody, (A) MMP-13, (B) TIMP-1, and (C) Collagen type II. (D-F) As result of IHC staining, percentage of positive staining area per total skin area was calculated using"
In Figure 7, we could only observe circles instead ***/**/*
Ethical concerns:
Some aspects related to the experimental work remain to be clarified, namely the definition of the human endpoints and the response during the experiment (which probably reached considerable levels of suffering - pain, anguish and stress in the animals used).
Author Response
We appreciate reviewers’ valuable comments and the opportunity to re-submit the manuscript, entitled “Effects of SKCPT on Osteoarthritis in Beagle Meniscectomy and Cranial Cruciate Ligament Transection Models”. Reviewers’ comments are insightful and we have revised the manuscript accordingly.
*The modified part is marked in red in the 'ijms-2610633_Revised manuscript 2'.
1 - The meniscectomy and cranial cruciate ligament transection treated Beagle dog model was used to investigate the efficacy of modified release SKI306X (SKCPT) formulation from Clematis mandshurica, Prunella vulgaris, and Trichosanthes kirilowii in managing arthritis. (the same at introduction section)
Response: Thank you for your valuable comment. The abstract can be considered as the beginning of the article that encompasses an overarching summary, including the content of the introduction. As it serves to introduce key concepts, the main content is reiterated. We kindly ask for your understanding in this regard.
2 - Repeated ideas
The group treated with 300 mg of Modified Release (MR) formulation demonstrated superior efficacy. Pro-inflammatory cytokines and matrix metalloproteinases were significantly lower in the SKCPT-treated groups than in the control group. This research found that the SKCPT reduces arthritic lesions and improves abnormal gait. The 300 mg MR formulation was more efficacious than others, suggesting a promising approach for managing OA symptoms and addressing disease pathogenesis.
Response: Thank you for your comment. As you pointed out, the repetitive content has been removed.
→ Modified section: Page 1
In the article "Quicke, J. G., Conaghan, P. G., Corp, N. & Peat, G. Osteoarthritis year in review 2021: epidemiology & therapy. Osteoarthr. Cartil. 30, 196–206 (2022)." I can't find any reference to the phrase: "In 2021, >22% of adults older than had knee OA, and it is estimated that over 500 million individuals are currently affected by OA worldwide.", cited by the article "Yao, Q.; Wu, X.; Tao, C.; Gong, W.; Chen, M.; Qu, M.; Zhong, Y.; He, T.; Chen, S.; Xiao, G., Osteoarthritis: pathogenic signaling pathways and therapeutic targets. Signal transduction and targeted therapy 2023, 8, (1), 56." which you in turn mentioned.
Response: Thank you for your comment. To avoid redundancy and plagiarism issues, this sentence was written by referencing papers 2, 3, and 4 when constructing the entire statement.
4.3
Clematis mandshurica, Prunella vulgaris, and Trichosanthes kirilowii, powdered extracts. as in the abstract
Table 1. Magnesium stearate
Response: Thank you for your detailed feedback. We have made the necessary corrections to the mentioned section in capital letters.
→ Modified section: Page 12
"The paraffin-embedded tissue slides were stained with each dye after deparaffinization and rehydration processes. Quantification of cartilage structure, chondrocyte pathology, proteoglycan staining, lining cell characteristics, cell infiltration characteristics, and hyperplasia characteristics was performed based on the results of (A) H&E and (B) Safranin-O staining. These parameters were graded according to their severity, and compared among groups, with the results expressed as (C) the OARSI score."
This information is provided in the Material and Methods section and is not expected in a figure legend. The captions should take the opportunity to describe the differences found in the various slides, which clearly do not appear in the captions of this work. Once again, scale bars are requested to be placed on photographs of all histological preparations.
the same for Figure 6 "After deparaffinization, rehydration, antigen retrieval, and blocking processes, the paraffin-embedded tissue slides were stained with each primary antibody, (A) MMP-13, (B) TIMP-1, and (C) Collagen type II. (D-F) As result of IHC staining, percentage of positive staining area per total skin area was calculated using"
Response: We appreciate your comment. In the case of the ‘Figure legend’ part, we modified it as you mentioned. In Figure 5 legend, the part was deleted, and in Figure 6 legend, unnecessary parts were removed and modified.
As previously mentioned, due to the inclusion of multiple images in one figure, adding annotations or bars caused the image to break, making verification difficult. Additionally, including all the information you mentioned in one stained image is challenging due to the variety of evaluations made from a single image, and it also makes comprehension difficult. In this situation, we kindly ask for your understanding that attaching such is problematic. If attachment is necessary, we would need to request documents from the ‘Committee of Knotus’ to provide better-quality images, and thus, additional time will be required. Could we possibly request a bit more time, if you would permit?
→ Modified section: Page 7 and 8
In Figure 7, we could only observe circles instead ***/**/*
Response: Thank you for your meticulous comments. It is indeed written as ***/**/*, but due to the resolution, it seems to appear as a circle.
Some aspects related to the experimental work remain to be clarified, namely the definition of the human endpoints and the response during the experiment (which probably reached considerable levels of suffering - pain, anguish and stress in the animals used).
Response: We sincerely appreciate your concerns regarding the ethical treatment of animals used in our research. In our study, we adhered strictly to the ethical guidelines and defined humane endpoints. These criteria include signs of severe pain, distress, inability to access food or water, and other significant health deteriorations. Any animal reaching these endpoints was humanely euthanized to prevent undue suffering. Throughout the experimental duration, animals were closely monitored daily for signs of distress, including pain, anguish, and stress. Immediate actions were taken if any signs of significant distress were observed. We understand the importance of transparency in these matters and tried to keep strict criteria.

Round 3
Reviewer 4 Report
Dear authors,
There remain some questions that have not been adequately clarified.
Abstract
The meniscectomy and cranial cruciate ligament transection treated Beagle dog model was used to investigate the efficacy of modified release SKI306X
What is at stake in this sentence is that Beagle dogs were subjected to experimental surgery to create an induced clinical situation that mimics a certain disease. Therefore, these animals were not treated for any pre-existing clinical situation, so these surgical procedures aimed to create an animal model and this is the notion that I consider should be included in the sentence.
4.3. Sample preparation and evaluation of drug efficacy according to dissolution pattern
SKCPT tablets (SK Chemicals Co., Ltd., Seongnam, Korea), consisting of the exact composition of API in various concentrations and formulations, were formulated from Clematis mandshurica, Prunella vulgaris, and Trichosanthes kirilowii powdered extracts.
Referred to uniformly as SKI306X in this study, this study utilizes refined extracts of Clematis mandshurica, Prunella vulgaris, and Trichosanthes kirilowii which exhibit diverse release patterns and are formulated with 30% ethanol extract (40→1).
Author Response
We appreciate reviewers’ valuable comments and the opportunity to re-submit the manuscript, entitled “Effects of SKCPT on Osteoarthritis in Beagle Meniscectomy and Cranial Cruciate Ligament Transection Models”. Reviewers’ comments are insightful and we have revised the manuscript accordingly.
*The modified part is marked in red in the 'ijms-2610633_Revised manuscript 3'.
The meniscectomy and cranial cruciate ligament transection treated Beagle dog model was used to investigate the efficacy of modified release SKI306X
What is at stake in this sentence is that Beagle dogs were subjected to experimental surgery to create an induced clinical situation that mimics a certain disease. Therefore, these animals were not treated for any pre-existing clinical situation, so these surgical procedures aimed to create an animal model and this is the notion that I consider should be included in the sentence.
Response: Thank you for your valuable comment. As your instructions, we have added an additional description of the experimental model in the first sentence of the '4.2. Meniscectomy and Cranial Cruciate Ligament Resection (CCLx)' section in the Materials and Methods.
→ Modified section: Page 11
→ Revised manuscript: Beagle dogs underwent experimental surgery to mimic a specific disease, OA, without prior treatment, for research purposes.
4.3. Sample preparation and evaluation of drug efficacy according to dissolution pattern
SKCPT tablets (SK Chemicals Co., Ltd., Seongnam, Korea), consisting of the exact composition of API in various concentrations and formulations, were formulated from Clematis mandshurica, Prunella vulgaris, and Trichosanthes kirilowii powdered extracts.
Referred to uniformly as SKI306X in this study, this study utilizes refined extracts of Clematis mandshurica, Prunella vulgaris, and Trichosanthes kirilowii which exhibit diverse release patterns and are formulated with 30% ethanol extract (40→1).
Response: Thank you for your comment. As you pointed out, the repetitive content has been removed and modified.
→ Modified section: Page 12
→ Revised manuscript: This study utilizes refined extracts of SKI306X, which exhibit diverse release patterns, and composed with 30% ethanol extract (40→1).
